# Dynamic Interplay of Online Risk and Resilience in Adolescence (DIORA): a protocol for a 12-month prospective observational study testing the associations among digital activity, affective and cognitive reactions and depression symptoms in a community sample of UK adolescents

Katarzyna Kostyrka-Allchorne [1] , Mariya Stoilova [2] , Jake Bourgaize,[3]
Aja Murray [4] , Eliz Azeri,[5] Chris Hollis,[6] Ellen Townsend [7] ,
Sonia Livingstone [8] , Edmund Sonuga-Barke,[9] on behalf of the Digital Youth Research Team

SL and ES-B are joint senior authors.

For numbered affiliations see end of article.

**Correspondence to**
Professor Edmund Sonuga-Barke;
edmund.sonuga-barke@kcl.ac.uk

## ABSTRACT

**Introduction** The impact of digital activity on adolescent mental health has been difficult to assess because of methodological limitations and a lack of strong theory. *Dynamic Interplay of Online Risk and Resilience in Adolescence* (DIORA) is a longitudinal study designed to address these core limitations and tease apart the reciprocal influences linking digital activity and depression symptoms (hereafter 'depression') over 12 months in middle adolescence. This study will examine whether negative affective and cognitive reactions evoked by risky digital activities increase depression. It will additionally examine whether protective characteristics (eg, self-efficacy) moderate the associations between digital activity and depression. DIORA will also explore the reverse pathways between digital activity and depression, namely whether depression exacerbates negative affective and cognitive reactions and, in turn, increase risky digital activities or, further, whether risks can be mitigated through active management of digital activity and/or reactions that it evokes. Finally, the study will examine whether the effects of digital activity observed for depression contrast with those observed for well-being.

**Methods and analysis** This is a prospective observational study with three assessment points: baseline (T1), 6 months (T2) and 12 months (T3). We aim to recruit a minimum of 276 adolescents aged between 13 and 14 years from secondary schools in the UK and 1 parent/caregiver/guardian (hereafter, 'parent') for each adolescent. Study questionnaires will be completed online. We will fit a range of models to examine the direct and indirect associations among digital activity, the reactions it evokes, depression and wellbeing, and individual and contextual mediators and moderators drawing on the structural equation modelling framework.

**Ethics and dissemination** The study was approved by the London School of Economics and Political Science Research Ethics Committee, reference number 249287. The results will be published in peer-reviewed scientific journals and disseminated through presentations, posters and blogs.

## STRENGTHS AND LIMITATIONS OF THIS STUDY

⇒ The Dynamic Interplay of Online Risk and Resilience in Adolescence (DIORA) study adopts a prospective longitudinal design to test the reciprocal relationships between digital activity and depression symptoms in a community sample of UK adolescents.

⇒ We will improve on simple measures of 'screen time' or 'amount of social media use' through use of the Digital Activities and Feelings Inventory (DAFI), which was codesigned with adolescents to identify and distinguish a range of digital activities, including risky activities, and the cognitive and affective reactions they evoke.

⇒ Furthermore, the study will assess the impact of cognitive and affective responses to digital activities as a predictor of depression symptoms.

⇒ Since the meaning and impact of digital activity are likely to vary across adolescence, the study focuses on mid-adolescence (13–14 years old at baseline), a period when the likelihood of depression rises.

⇒ It will not be feasible to implement a reliable objective measure of adolescents' digital activity and mental health, although we will collect proxy measures from parents.

## INTRODUCTION

Adolescence is a unique period of substantial biological, psychological and social changes. It is also a period that is associated with an increased likelihood of developing mental health problems, especially depression.[1 2] In Western countries, adolescent mental health problems have been steadily increasing over the past three decades. In the UK, for adolescents aged 17 to 19 years, rates of probable mental disorders rose from 1 in 10 in 2017 to 1 in 4 by 2022.[3] In parallel, there has been a consistent increase in the ownership of digital devices (eg, smartphones, laptops, game consoles). For example, in the UK, 98% of adolescents aged 12–15 years own a mobile phone.[4] Inevitably, digital activity has become an integral part of adolescent life, with many adolescents reporting to be almost constantly online.[5 6] This has led to concerns that engaging in digital activity may be harmful to adolescent mental health and psychological wellbeing.[7]

However, research findings are nuanced,[8] with small effect sizes,[9] and subject to measurement problems[10] that require careful contextualisation.[11] Systematic reviews of the literature on the link between digital activity and mental health reveal mixed and inconsistent results.[7 12–15] Specifically, results vary depending on the type of online risk[16] and users' mood and susceptibility,[17] as well as on their vulnerability and available support.[18] At the same time, potential benefits of digital activity have been highlighted, including opportunities for social connection, support and entertainment[19 20] and greater control over how one selects and engages with the external environment.[21] Indeed, research identifies the benefits of digital activity, though few studies attempt or find links with robust positive mental health or wellbeing measures.[22 23] Literature reviews often call for more specificity in the construction of measures and research designs.[24 25]

Finding adequate measures of digital activity has proved particularly challenging, with many studies using a simple measure of 'screen time' or 'social media use'.[26] This is important because different digital activities might be linked to different mental health outcomes.[27] That is, some digital activities are more likely to contribute to poor mental health than others. For example, seeing violent or self-harm content, being treated in a hurtful way or comparing oneself adversely to others are likely to confer risks for mental health and hereafter will be referred to as 'risky digital activities'.[28–30] By contrast, entertainment activities, such as playing a digital game or having fun with other people online, are more likely to be either neutral or perhaps even protective for mental health. Mental health outcome measures can also be problematic, with some studies failing to distinguish between different mental health conditions, even though the effects may differ for anxiety and depression[31] and the effects for wellbeing may not simply be the inverse of those for depression.[32] Regarding research design, prior studies have often mixed participants of different ages or developmental stages, although the effects may be developmentally specific.[33] However, the most criticised aspect of prior designs is reliance on cross-sectional data even though it is vital to clarify the causal direction: does digital activity increase the likelihood of developing mental health problems or does mental health determine digital activity?[34] Related to this issue of causality is the need to locate the effects of digital activity in the context of offline risk and protective factors. For example, controlling for confounding risks may change the digital activity–mental health relationship.[35–37] Other potential mediating or moderating mechanisms may also drive or alter the relationship between digital activity and mental health.[38]

## AIMS AND OBJECTIVES

To address these limitations, the present study aims to examine the reciprocal longitudinal direct and indirect pathways between adolescent mental health and wellbeing and digital activity.

The study was designed to test a recently proposed cognitive-affective theory of the complex and reciprocal relationships between digital activity and depression.[39] In this study, we propose three core hypotheses.

First, risky digital activity impacts on depression through the frequency and persistence of the negative reactions it evokes in individuals. These encompass both negative affect (eg, anger, annoyance) and negative cognitions (eg, 'I am stupid', 'I am inferior').

Second, depression will, in turn, exacerbate risky digital activities (ie, leading individuals to engage in activities that contribute to poor mental health) or the amount of negative affective and cognitive reactions it evokes, creating the potential for destructive cycles that may alter the course of development.

Third, individuals who recognise the negative impact of digital activity on their mental health might be able to mitigate it by either reducing their risky digital activities and/or by modifying their reactions to it. This, in turn, will lead to better mental health outcomes.

The *primary objective* of this study is to examine whether digital activity predicts an increase in depression symptoms over a 12-month period in middle adolescence (ie, age 13–14 years), the developmental period when depression likelihood is beginning to increase.[40]

Secondary objectives are to examine:

1. Whether depression is prospectively associated with risky digital activities over a 12-month period.
2. Whether the pathway from digital activity to depression is mediated by affective and cognitive reactions evoked by digital activity.
3. Whether depression feeds back to alter patterns of digital activity and the reactions it evokes, reciprocally.
4. Whether personal resilience and other known risk and protective factors moderate the pathway from digital activity to depression and vice versa.
5. Whether effects observed for wellbeing are the inverse of those observed for depression.
6. Explore whether an individual's reflections about the impact that their digital activity is having on their

mental health leads to their modifying their pattern of digital activity and/or their reactions to it.

We will also explore which factors predict high-risk digital activities and what proportion of adolescents are concerned about the impact of digital activity on their lives and mental health. What proportion attempt to modify their activity in light of this? What approaches do they use?

## METHODS AND ANALYSIS
### Study design
This is a longitudinal observational study with measures taken at baseline (T1), 6 months (T3) and 12 months (T3) post baseline.

### Study setting
All study procedures (consent form and questionnaire completion) will be completed remotely using Qualtrics. Participants will either complete the study measures at school or use their own devices connected to the internet in their preferred setting.

### Participant selection and eligibility criteria
Participants in the study will be adolescents and their parents. Adolescents will be recruited from secondary schools in the United Kingdom (UK). Each adolescent will nominate a parent (or caregiver/guardian), who will also be invited to participate. To ensure participants represent a wide socioeconomic spectrum, we will approach a range of schools including mainstream and alternative provision schools, either state funded or independent. To identify schools, we will use contacts in our professional networks, as well as information from the Department for Education. We will continue approaching schools with a request to support recruitment until sufficient adolescents have consented to the study. A member of the research team will meet with each school's nominated representative to discuss the study.

#### Inclusion criteria
1. Adolescents attending a secondary school in England, Wales or Scotland.
2. The adolescents are aged 13–14 years old at the time of enrolment (typically, Year 9 in England and Wales or S3 in Scotland).
3. Adolescents with access to a digital device connected to the internet (eg, owned or shared at home, or a publicly shared device).
   There are no specific inclusion criteria for parents.

#### Exclusion criteria
Adolescents will be excluded if they are unable to complete assessments independently.

### Sample size justification
We judged the minimum clinically meaningful effect size of interest to detect, based on potential public health significance, to be r=0.2. Our cross-sectional pilot study

(publication forthcoming) found correlations with much larger effect sizes than this between proposed predictors and outcomes—suggesting that this effect size is obtainable. An a priori biserial normal model was used to calculate the required sample size.[41] To achieve 80% power for an alpha of 0.05, this power analysis indicated a sample size of 193 is needed. Assuming a conservative 70% retention at 12-month follow-up, we will recruit a minimum of n=276 adolescents; however, we will aim for the sample to be larger to allow sufficient power for moderation/mediation analyses.

### Recruitment and follow-up procedures
Researchers will work with schools to inform adolescents and parents about the study and to provide them with opportunities to ask questions. This may involve visiting schools to talk about the study during assemblies/classes, asking schools to display posters and share flyers with eligible students and circulating study information via the school's newsletter, website and social media.

Schools will also be asked to share a digital recruitment pack with each adolescent, which will include study promotion materials (eg, a flyer) and information sheets for adolescents and parents. The information sheet will include a link (eg, a QR code) to an online consent form for the adolescent.

The school will also choose one of the two approaches regarding parental consent, that is, the opt-in and opt-out options. Under the opt-in approach, adolescents will not be able to participate unless the parent (or legal guardian) actively consents to their child's participation by filling in an online consent form. Under the opt-out approach, adolescents will be asked to seek verbal consent from their parent (or legal guardian) and will be able to participate unless their parent (or legal guardian) explicitly chooses for their child not to participate in the study. Under both consent approaches, adolescents and parents will need to consent to their own participation.

Adolescents and their parents will provide informed consent online. Following consent, adolescents and parents will receive a notification via email and/or text message to complete the battery of T1 questionnaires online. Each assessment window will close after 3 weeks of opening. A notification to complete assessments will be sent via email and/or text message on the day that the assessment is due to be completed.

Each adolescent will receive a £15 voucher for completing the questionnaires at each time point and a 'bonus' of £15 for completing assessments at all time points (a total of £60 for those who complete the whole study). This compensation structure was suggested by the Dynamic Interplay of Online Risk and Resilience in Adolescence (DIORA) youth panel as the most optimal to maintain adolescents' engagement in the study. Parents will be entered into a prize draw. Each school facilitating recruitment will receive between £100 and £250 to thank them for their support (amount depending on the level of support given to the study's recruitment). Other

non-monetary incentives (eg, psychology talks, school-level feedback, etc) will be negotiated with individual schools.

### Participant timeline
Adolescent and parent measures will be completed at T1, T2 and T3. A detailed schedule of assessments is presented in online supplemental table 1.

### Outcomes
#### Adolescent-reported outcomes

#### Mental health and wellbeing
Adolescents will report their depression and anxiety symptoms using the *Revised Child Anxiety and Depression Scale 25—Youth Version* (RCADS-25;[42]). Adolescents will rate how often each statement applies to them on a 4-point scale (0=never, 3=always). The subscale score is derived by summing individual item scores. Higher scores on the subscale indicate greater symptoms of depression and/or anxiety. Conduct problems and attention-deficit hyperactivity symptoms will be measured with the respective subscales of the *Strengths and Difficulties Questionnaire.*[43] In this 25-item questionnaire, adolescents will be provided with descriptions of personal characteristics and asked to indicate whether the description is true of themselves on a 3-point rating (0='not true' to 2='certainly true'). After reverse coding, individual item scores are summed to derive an overall subscale score, where the higher score represents greater severity of symptoms. Wellbeing will be measured with the *Warwick-Edinburgh Mental Wellbeing Scale* (WEMWBS;[44]). It assesses the presence of positive thoughts and feelings, which relate to one's general functioning. All statements are positively worded and are rated on a 5-point rating scale (1='none of the time' to 5='all of the time'). Individual item scores are summed to calculate the total score.

#### Digital activity
Screen time will be measured with a single question: 'On an ordinary school day, about how long do you send on your phone or the internet (not counting time for schoolwork)?' Responses are provided on a 9-point scale ('little to no time' to 'about 7 hours'). Adolescents will also complete a detailed assessment of their digital activity using the recently developed *My Online Activity* subscale of the *Digital Activity and Feelings Inventory* (DAFI). It measures four distinct dimensions of digital activity: high-risk content (seven items; for example, 'I watched some sad or dark content'), high-risk conduct (eight items; eg, 'I saw people talk about or show ways of being very thin'), entertainment (three items; eg, 'I played games with others') and social use (four items; eg, 'I like or shared other people's posts'). Adolescents will rate how often they did or experienced each item in the past 2 weeks on a 5-point scale (0='never' to 4='at least every day'). Reactions evoked by digital activity will be measured with the *My Feelings Online* subscale of the DAFI, which includes 10 positive and 10 negative feelings that encompass both

general emotional reactions (eg, 'stressed', 'calm') and those that specifically pertain to the self (eg, 'insecure', 'attractive', 'loved'). Adolescents will rate how often they experienced each item in relation to being online in the past 2 weeks on a 5-point scale (0='never' to 4='at least every day'). Information about the validation of this measure will be reported in the forthcoming publication. The impact of online activities on daily life and management behaviours will be measured by a second new measure: the *My Life Online* (MYLO) questionnaire. Section 1 (nine items), asks adolescents about how much their life was impacted by their digital activity (eg, 'I missed meals', 'I had difficulties at school/college/university') rated on a scale of 0='never' to 4='at least every day' and whether they were concerned about this (eg, 'I worried how being online affected my mental health') rated on a scale of 0='not at all' to 4='very much'. In section two of MYLO adolescents (14 items) will rate the extent they tried to reduce the negative effects of their digital activity (eg, 'Took a break from social media', 'Removed or blocked accounts'). Responses are provided on a scale ranging from 0='never' to 4='at least every day'.

#### Individual characteristics
Finally, we will measure the following individual characteristics: psychological resilience, social comparison, self-efficacy, emotion regulation and social desirability. We will also measure the following contextual factors: offline events and experiences, loneliness, social support and the presence of significant life events. A demographic questionnaire will ask for information about adolescents' age, biological sex and gender identity, ethnicity, eligibility for free school meals and perceived family affluence. A detailed description of all the scales and their administration is provided in online supplemental table 1.

#### Parent-reported outcomes
#### Mental Health
Parents will report their child's depression and anxiety symptoms on the respective subscales of the RCADS-25. Adolescent psychological wellbeing will be measured with an adapted version of the WEMWBS. Parents will also complete a parent-report version of MYLO to rate how much being online has affected their child's everyday life and will rate perceived positive and negative impacts of their child's online activity on their mental health and relationships. They will also report on their relationship with their child on the Child–Parent Relationship Scale—Short Form,[45] which includes two subscales: the 7-item closeness subscale scale and the 8-item conflict subscale. Parents will rate how much each statement applies to their relationship with their child on a 5-point rating scale ranging from 'definitely does not apply'=1 to 'definitely applies'=5.

Parental depression will be measured with the Patient Health Questionnaire,[46] which assesses depression symptoms using a 4-point scale ranging from 'not at all'=0 to 'nearly every day'=3. Individual item scores are summed to

derive an overall score—a higher overall score represents more severe depression symptoms. Parental wellbeing will be measured with the WEMWBS.

### Demographics

To obtain demographic information, parents will provide information about their own ethnicity, education and employment status, as well as report the occupation of the main earner in the family. The latter will be categorised using the nine major groups from the Office for National Statistics Standard Occupational Classification Hierarchy.[47]

### General procedure
#### Data collection methods

Data will be collected online using Qualtrics—a secure, dedicated survey platform. Each consenting participant (ie, adolescent and parent) will receive a link to the relevant T1 questionnaires. If a participant does not start the T1 questionnaires within 3 weeks, it will be assumed they do not wish to continue and will not be invited to complete follow-up assessments.

Participants who completed T1 questionnaires will be sent two follow-up assessment invitations (T2 and T3) via text message or email. At each time point, they will have 3 weeks to start completing the questionnaires and 7 days to finish the questionnaires once started. Up to three reminders will be sent to those who have not completed the assessment.

### Withdrawals

Participants will be free to withdraw at any point in the study by either ignoring invitations to complete the online questionnaires or by emailing the research team. Data collected before withdrawal will be included in the analysis.

### Public patient involvement and engagement: youth panel

The overall Digital Youth programme research priorities, ethical and design considerations were considered at the prefunding stage through the consultation with the Medical Research Council Adolescence, Mental Health & Developing Mind Engagement Award Young Person Advisory Group (now Sprouting Minds), a youth panel which informs and supports Digital Youth researchers through cofacilitation and collaboration.

In addition, DIORA has a dedicated youth advisory panel comprising adolescents aged 13–17 years, which meets online approximately every 2 months. The DIORA youth panel played a key role in the development of the DAFI to ensure that the list of digital activities and reactions they induce were relevant to adolescents. The panel also reviewed and advised on current study plans and procedures and participant-facing documents to ensure these are acceptable and inclusive. For example, the panel helped to short-list digital activities that are relevant to contemporary adolescents and advised that recruitment occurs through schools to increase the perceived legitimacy of research (also because students may not trust or

know the academic institutions that are conducting the study). We will continue working with both Sprouting Minds and the DIORA youth panel throughout this project and future activities will include reviewing the progress of the longitudinal study, interpreting findings and involvement in results dissemination.

### Analysis plan

We will examine information about the proportion of missing data and its patterns. Summary statistics will be used to describe demographic information and provide information about the level of psychopathology symptoms and digital activity in the adolescent sample.

We will fit a range of models to examine the associations among digital activity, depression and wellbeing drawing on the structural equation modelling framework, which provides the flexibility to implement a diversity of models which map to the study's objectives. For example, (random intercept) cross-lagged panel models will be used to examine (potentially reciprocal) directional relations over time between digital activity and depression; growth curve models will be used to examine trajectories over time, their individual differences, and predictors of these individual differences; and mediating paths can be added to models to examine indirect effects or 'mechanistic pathways'. Missing data will be dealt with using methods such as multiple imputation, full information maximum likelihood estimation and Bayesian estimation, but the specific approach will depend on the model, reflecting the greater compatibility of some approaches with certain modelling and estimation frameworks. Further details of the analyses will be specified in the statistical analysis plan.

## ETHICS AND DISSEMINATION
### Ethics approval and safety consideration

The study was approved by the London School of Economics Research Ethics Committee, reference number 249287. Any future protocol amendments will be documented and submitted for ethical approval prior to implementation.

Given the observational design the risk of harm to participants is minimal. There is a small likelihood that completing study questionnaires may result in new concerns related to parent or child mental health, digital activity, relationships or affective and cognitive reactions. This risk is clearly explained through the information sheet. All questionnaires/procedures have been reviewed by the DIORA youth panel to ensure sensitivity and acceptability to adolescents and their parents.

All participants will give written (electronic) informed consent for their participation. For adolescents recruited via the opt-in approach, we will also seek written parental informed consent (for details, see 'Recruitment and follow-up procedures'). Participants will be able to skip any question they do not want to answer, and all participants will be provided with information on seeking mental

health support and contacts for national crisis services at the start and end of each assessment battery.

## Dissemination

Results will be published in a peer-reviewed scientific journal and included in the funder report. Authorship will be determined according to the Digital Youth's publication policy developed on the basis of the Committee on Publication Ethics standards based on individual contributions. Findings will be disseminated through presentations at conferences and seminars in the UK and overseas. There will also be a general dissemination through the Digital Youth and institutional websites, social media and stakeholder engagement activities.

## Study oversight

Overall programme operational oversight and progress monitoring is provided by the Digital Youth leadership team. This group comprises the programme principal investigators (CH and ET), work package leads, programme manager and members of Sprouting Minds. Overall scientific oversight is provided by the Scientific Advisory Board, which reviews and approves study protocols and receives progress updates. This comprises international academic experts representing disciplines relevant to the programme (eg, adolescent mental health, computer science, digital psychological interventions).

## Study status

The study recruitment started in October 2023 with data collection planned to be completed in March 2025.

**Author affiliations**
[1]Department of Psychology, School of Biological and Behavioural Sciences, Queen Mary University of London, London, UK
[2]Department of Media and Communications, The London School of Economics and Political Science, London, UK
[3]Department of Child & Adolescent Psychiatry, King's College London, London, UK
[4]Department of Psychology, The University of Edinburgh, Edinburgh, UK
[5]Department of Psychology, King's College London, London, UK
[6]NIHR MindTech MedTech Cooperative and NIHR Nottingham Biomedical Research Centre, School of Medicine, Division of Psychiatry, University of Nottingham, Nottingham, UK
[7]School of Psychology, University of Nottingham, Nottingham, UK
[8]The London School of Economics and Political Science, London, UK
[9]Department of Child & Adolescent Psychiatry, King's College London, London, UK

**Correction notice** This article has been corrected since it was published. Data availability statement has been added.

**Contributors** The guarantors of the study are SL and ES-B; they accept full responsibility for the conduct of the study, will have access to the data, and will control the decision to publish. SL, ES-B, MS, JB, AM and KK-A contributed to the design of the study. AJ, ES-B and SL contributed to the analysis plans. MS and JB prepared the protocol and other study documentation for ethical approval. KK-A and MS prepared the first draft of the manuscript for publication. EA and JB organised the activities of the DIORA youth panel and liaised with Sprouting Minds. All authors contributed to drafting and revising the manuscript and approved the final version to be submitted for publication.

**Funding** The authors (MS, KK-A, JB, EA, CH, ET, SL and ES-B) acknowledge the support of the UK Research and Innovation (UKRI) Digital Youth Programme award (Medical Research Council (MRC) as part of a large multi-institutional programme of research called Digital Youth https://digitalyouth.ac.uk (funder reference MR/ W002450/1; chief investigators: CH and ET, University of Nottingham), which is part of the AHRC/ESRC/MRC Adolescence, Mental Health and the Developing Mind programme. The project funding covers all costs in relation to the study, including but not limited to, higher education institutions' costs, equipment and the youth panel costs. The funding covers the period from 1 September 2021 to 31 August 2025. This study is in part funded by the National Institute for Health and Care Research (NIHR) Maudsley Biomedical Research Centre (BRC). The views expressed are those of the author(s) and not necessarily those of the NIHR or the Department of Health and Social Care."

**Competing interests** None declared.

**Patient and public involvement** Patients and/or the public were involved in the design, or conduct, or reporting, or dissemination plans of this research. Refer to the Methods and analysis section for further details.

**Patient consent for publication** Not applicable.

**Provenance and peer review** Not commissioned; externally peer reviewed.

**Data availability statement** Data sharing not applicable as no datasets generated and/or analysed for this study.

**ORCID iDs**
Katarzyna Kostyrka-Allchorne https://orcid.org/0000-0002-0789-2449
Mariya Stoilova https://orcid.org/0000-0001-9601-7146
Aja Murray https://orcid.org/0000-0002-9068-3188
Ellen Townsend https://orcid.org/0000-0002-4677-5958
Sonia Livingstone https://orcid.org/0000-0002-3248-9862

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
