## [Reviewer comments · BMJ Open]

ARTICLE DETAILS

TITLE (PROVISIONAL)	Dynamic Interplay of Online Risk and Resilience in Adolescence (DIORA): a protocol for a 12-month prospective observational study testing the associations between digital activity, affective and cognitive reactions and depression symptoms in a community sample of UK adolescents
AUTHORS	Kostyrka-Allchorne, Katarzyna; Stoilova, Mariya; Bourgaize, Jake; Murray, Aja; Azeri, Eliz; Hollis, Chris; Townsend, Ellen; Livingstone, Sonia; Sonuga-Barke, Edmund; Youth, Digital

VERSION 1 - REVIEW

REVIEWER NAME	Mayer, Gwendolyn
REVIEWER AFFILIATION	University Hospital Heidelberg, Internal Medicine and Psychosomatics
REVIEWER CONFLICT OF INTEREST	I have no conflicts of interests.
DATE REVIEW RETURNED	14-Mar-2024

GENERAL COMMENTS	The presented study protocol aims at examining whether negative affective and cognitive reactions evoked by risky digital activity increase depressive symptoms. The protocol is well written and presented in a structured way. Few points that might improve the paper: 1. The abstract introduction: is resilience only a “characteristic”? Moreover, does it moderate the association between digital activity and depression or between risky behaviour and depression (see comment 4)2. In general the terms “digital activity” and “risky activities” are often used interchangeably, I recommend a more precise term that should be used consistently3. Shorten the text used in the bullet points4. In general, I miss a distinct definition of risks. In the hypotheses, digital risks are described as “leading individuals to engage in more negative digital activity”, later on you mention depression risks, which is confusing. Negative digital activity is an adult interpretation of adolescent behaviour and might not be perceived negative by young people. Moreover, risky behaviour is an inherent part of being adolescent. So, what is the exact aim in this study? Internet gaming risks (financial), being negatively influenced by social media (body image, e.g. anorexia), losing contact to reality (what ever is meant by this)? And in what way might this relate to “offline-risks”? (see my comment 8)5. Literature reviews often call for more specificity in the construction of measures and research designs -> provide a source6. A subheading with Aims and objectives would be useful7. Methods: A missing measure is the online behaviour of parents and their attitudes.
--

	8. Did you consider to ask for risky behaviour outside the internet (drugs, delinquency). 9. Handling missing data – in an online survey you have the opportunity to define parts as mandatory, has this been done? 10. How do you manage drop-outs? 11. How do you deal with self-reported self-harm/suicidality? This is an ethical question, what if PHQ shows increased values? 12. I strongly recommend to control for socio-economic background, choose a wide variety of schools 13. Would a network analysis at the level of symptoms and online behaviour be useful? Thank you and good success with this interesting work!
--	---

REVIEWER NAME	Goodyear, Victoria
REVIEWER AFFILIATION	University of Birmingham
REVIEWER CONFLICT OF INTEREST	None
DATE REVIEW RETURNED	11-Apr-2024

GENERAL COMMENTS	This study protocol is through, detailed and covers all expected components.
--

VERSION 1 – AUTHOR RESPONSE

Comments by Reviewer 1

1. The abstract introduction: is resilience only a “characteristic”? Moreover, does it moderate the association between digital activity and depression or between risky behaviour and depression (see comment 4).

Response: Thank you for pointing this out, we have replaced ‘resilience’ with ‘self-efficacy’. Regarding the role of resilience, we have referenced an earlier paper by the team (see reference 36).

2. In general the terms “digital activity” and “risky activities” are often used interchangeably, I recommend a more precise term that should be used consistently –

Response: We have made edits to make it clearer that we intentionally use the two different terms, and they are not interchangeable. Specifically, we have aimed to make it clearer that “Digital activity” is the more neutral term that refers to all online activities and not all of these are risky. This is distinguished from “Risky activities”, which refers to a sub-set of digital activities that may be risky for poor mental health (for example accessing content that is related to self-harm or making unfavourable social comparisons). Thus, while we retained the two terms, we have read the text carefully again with the reviewer’s comment in mind and have clarified the distinction in the text: *“That is, some digital activities are more likely to contribute to poor mental health than others. For example, seeing violent or self-harm content, being treated in a hurtful way, or comparing oneself adversely to others are likely to confer risks for mental health and hereafter will be referred to as ‘risky digital activities’. In contrast, entertainment activities, such as playing a digital game or having fun with other people online, are likely to be either neutral or perhaps even protective for mental health.”*

3. Shorten the text used in the bullet points.

Response: We have revised the bullets, also in the light of the editor’s comment.

4. In general, I miss a distinct definition of risks. In the hypotheses, digital risks are described as “leading individuals to engage in more negative digital activity”, later on you mention depression risks, which is confusing. Negative digital activity is an adult interpretation of adolescent behaviour and might not be perceived negative by young people. Moreover, risky behaviour is an inherent part of being adolescent. So, what is the exact aim in this study? Internet gaming risks (financial), being negatively influenced by social media (body image, e.g. anorexia), losing contact to reality (what ever is meant by this)? And in what way might this relate to “offline-risks”? (see my comment 8).

Response: We absolutely agree with the reviewer that adults and children perceive risks differently and that it's important to distinguish between risks and harms (which our study aims to do). We have added early on (in the introduction) several key references - on the classification and meaning of online risks that we build on and on the effects on online risks on different aspects of mental health.

We have also clarified the meaning of risk: *“different digital activities might be linked to different mental health outcomes (27). That is, some digital activities are more likely to contribute to poor mental health than others. For example, seeing violent or self-harm content, being treated in a hurtful way, or comparing oneself adversely to others are likely to confer risks for mental health and hereafter will be referred to as ‘risky digital activities’ (28, 29, 30). By contrast, entertainment activities, such as playing a digital game or having fun with other people online, are likely to be either neutral or perhaps even protective for mental health.”*

Finally, we revised our language regarding risk throughout the protocol. We have also revised the sentence “leading individuals to engage in more negative digital activity” and “Risk of depression” has been replaced by “likelihood of depression” to avoid confusion as this does not refer to online risks. We hope that this helps to resolve any ambiguity.

5. Literature reviews often call for more specificity in the construction of measures and research designs -> provide a source.

Response: The relevant citations have been added in the text/reference list.

6. A subheading with Aims and objectives would be useful.

Response: Thank you for this helpful suggestion – we have created a suggested subheading.

7. Methods: A missing measure is the online behaviour of parents and their attitudes.

Response: This is a very good suggestion that could be implemented in a future study. Our current study includes parents with the aim of obtaining proxy measures on the impact of adolescents' digital activity and mental health. We kept the number of questions for parents to a minimum with the hope to increase completion rates as the parents will not receive incentives for the study.

8. Did you consider to ask for risky behaviour outside the internet (drugs, delinquency)
9. Response: Yes, we are measuring a range of offline risk and protective factors in this study. Risky offline activities, including drugs, missing school, etc, will be measured with the Personal Experiences in Everyday Life questionnaire. Please see Table 1 for the complete list and description of all study measures (due the word count limit we are unable to describe all these measures in the text).
10. Handling missing data – in an online survey you have the opportunity to define parts as mandatory, has this been done?

Response: Due to the sensitivity of the topic the participants can skip any question. This has now been clarified in the text.

11. How do you manage drop-outs?

Response: We discussed ways of managing drop out with our DIORA youth panel, which helped us decide how to compensate the participants to increase retention in the study. We now make this approach clear in the text: *“Each adolescent will receive a £15 voucher for completing the questionnaires at each time point and a ‘bonus’ of £15 for completing assessments at all timepoints. This compensation structure was suggested by the DIORA youth panel as the most optimal to maintain adolescents’ engagement in the study.”*

We have also clarified how we will deal with missing data in the analysis: *“Statistically, missing data will be dealt with using modern missing data methods that can provide unbiased parameter estimates under MAR, such as multiple imputation, Bayesian estimation, and full information maximum likelihood estimation. The specific missing data method employed will depend on the analysis as different methods may be better suited for different models (e.g., FIML for structural equation models and Bayesian methods when Bayesian estimation is used).”*

12. How do you deal with self-reported self-harm/suicidality? This is an ethical question, what if PHQ shows increased values?

Response: Thank you for raising this important point. We are unable to identify any participants who are in short-term crisis based on their PHQ response. However, our approach to safeguarding has been described in detail in our ethics application to the LSE Ethics Committee and has been approved. We are also following the safeguarding policies of LSE and KCL. In brief, risks are clearly explained through the information sheet. In addition, contact details for support organisations and websites that can provide support at times of mental health crisis are included within the online surveys. This information appears in the manuscript in section 6.2. ‘Ethics approval and safety consideration’.

13. I strongly recommend to control for socio-economic background, choose a wide variety of schools.

Response: Thank you for this great suggestion. This is an issue we have thought about extensively. We have a few measures to capture this including free school meals, parental education and occupation, and family affluence. Please see Table 1 for a complete range of measures. The details of the control variables will be specified in the future analysis plan.

14. Would a network analysis at the level of symptoms and online behaviour be useful?

Response: There are currently no specific plans for network analysis due to resource constraints, but this is an excellent suggestion for a secondary data analysis project.

Comments by reviewer 2: no revisions requested